# Synaptic cell adhesion molecule *Cdh6* identifies a class of sensory neurons with novel functions in colonic motility

**Julieta Gomez-Frittelli[1,2], Gabrielle Frederique Devienne[2,3], Lee Travis[4], Melinda A Kyloh[4], Xin Duan[5], Tim J Hibberd[4], Nick J Spencer[4], John R Huguenard[2,3], Julia A Kaltschmidt[2,6]\***

[1]Department of Chemical Engineering, Stanford University, Stanford, United States; [2]Wu Tsai Neurosciences Institute, Stanford University, Stanford, United States; [3]Department of Neurology & Neurological Sciences, Stanford University, Stanford, United States; [4]College of Medicine and Public Health, Flinders Health & Medical Research Institute, Flinders University, Adelaide, Australia; [5]Department of Ophthalmology, School of Medicine, University of California San Francisco, San Francisco, United States; [6]Department of Neurosurgery, Stanford University School of Medicine, Stanford, United States

**\*For correspondence:**
jukalts@stanford.edu

## eLife Assessment

This **important** study characterizes the molecular signatures and function of a type of enteric neuron (IPAN) in the mouse colon, identifying molecular markers (Cdh6 and Cdh8) for these cells. A battery of **compelling** and comprehensive experimental findings suggests data from other species are likely translatable to mice, bridging the abundant literature from humans and other mammals into this experimentally tractable animal model. This work will be of interest to scientists studying the motor control of the colon and more generally the enteric neuromuscular system.

**Abstract** Intrinsic sensory neurons are an essential part of the enteric nervous system (ENS) and play a crucial role in gastrointestinal tract motility and digestion. Neuronal subtypes in the ENS have been distinguished by their electrophysiological properties, morphology, and expression of characteristic markers, notably neurotransmitters and neuropeptides. Here, we investigated synaptic cell adhesion molecules as novel cell-type markers in the ENS. Our work identifies two type II classic cadherins, *Cdh6* and *Cdh8*, specific to sensory neurons in the mouse colon. We show that *Cdh6+* neurons demonstrate all other distinguishing classifications of enteric sensory neurons including marker expression of *Calcb* and *Nmu*, Dogiel type II morphology and AH-type electrophysiology and $I_H$ current. Optogenetic activation of *Cdh6+* sensory neurons in distal colon evokes retrograde colonic motor complexes (CMCs), while pharmacologic blockade of rhythmicity-associated current $I_H$ disrupts the spontaneous generation of CMCs. These findings provide the first demonstration of selective activation of a single neurochemical and functional class of enteric neurons and demonstrate a functional and critical role for sensory neurons in the generation of CMCs.

## Introduction

Sensory signaling within the gastrointestinal (GI) tract plays a critical role in the autonomous regulation of digestion. The GI tract is the only internal organ system containing its own sensory neurons. Intrinsic

primary afferent neurons (IPANs) detect relevant stimuli through chemo- and mechano-sensation and direct appropriate GI functions via downstream components of the enteric nervous system (ENS), including ascending and descending interneurons, and excitatory and inhibitory motor neurons (**Fung and Vanden Berghe, 2020**). These neuronal subtypes have begun to be distinguished morphologically, electrophysiologically, and by marker expression, classically especially of neurotransmitters (**Nurgali et al., 2004**; **Qu et al., 2008**), and together this information has provided an opening to characterize individual neuron subtype function within the GI tract.

Synaptic cell adhesion molecules define neuronal subtype connectivity within many regions of the CNS. Type II cadherins are a family of synaptic cell adhesion molecules with combinatorial expression in multiple neural circuits of the CNS, including retina, limbic, olivonuclear, and auditory projection systems (**Suzuki et al., 1997**; **Duan et al., 2018**; **Honjo et al., 2000**). Type II cadherins bind homophilically by expression of the same cadherin at both the pre- and post-synapse, which stabilizes developing synapses between correct partners while incorrect synapses are pruned away (**Yamagata et al., 2018**; **Basu et al., 2015**). Recent RNA-Seq studies of human and mouse ENS have identified synaptic cell adhesion molecules, including type II cadherins, expressed in enteric neuronal subtypes (**May-Zhang et al., 2021**; **Drokhlyansky et al., 2020**; **Morarach et al., 2021**). However, the specificity of their expression has yet to be harnessed to assess neuronal subtype-specific function in the ENS.

Here, we identify type II cadherin, *Cdh6*, as a novel marker for IPANs of the colonic ENS. We demonstrate the sensory identity of *Cdh6* neurons by immunohistochemical, morphological, and neurophysiological classification. *Cdh6* neurons express IPAN markers *Calcb* and *Nmu*. Sparse labeling of individual IPANs reveals they project mainly circumferentially and branch extensively in myenteric ganglia. Whole-cell patch-clamp recordings of sensory neurons in situ reveal action potential (AP) slow afterhyperpolarizations characteristic of IPANs, and hyperpolarization-activated cationic current ($I_H$), a rhythmicity indicator in thalamocortical and other systems (**Wahl-Schott and Biel, 2009**). Using a *Cdh6* genetic mouse model, we show that optogenetic activation of distal colon IPANs is sufficient to evoke retrograde colonic motor complexes (CMCs), while pharmacologic block of $I_H$ in IPANs disrupts colonic rhythmicity and reversibly abolishes spontaneous CMCs.

## Results

### Expression of the type II classic cadherin *Cdh6* in colonic IPANs

To identify cadherins expressed in enteric neuronal subtypes in mouse, we screened recently published RNA-Seq data (**Drokhlyansky et al., 2020**; **Morarach et al., 2021**) for classic type II cadherin expression. *Cdh6* and *Cdh8* appeared to be restricted to IPAN subsets in both small intestine and colon (**Drokhlyansky et al., 2020**; **Morarach et al., 2021**). *Cdh9* was previously identified in a separate population of IPANs in the small intestine (**May-Zhang et al., 2021**; **Drokhlyansky et al., 2020**; **Morarach et al., 2021**). We validated *Cdh6* and *Cdh8* expression by RNAscope in situ hybridization in the myenteric plexus, which contains the enteric motility circuitry. *Cdh6* mRNA was expressed in 14.7 ± 0.8% of myenteric neurons in small intestine (jejunum) and in 6.8 ± 0.3% of myenteric neurons in distal colon (mean ± SEM) (**Figure 1A–C**). *Cdh8* was almost exclusively co-expressed in *Cdh6+* neurons, although at a much lower level of detection (**Figure 1D–H**). We therefore focused our further analysis on *Cdh6+* neurons.

To confirm IPAN identity of *Cdh6+* neurons, we first established the differential expression of two putative and broadly used markers of IPANs, *Calcb* and *Nmu* (**Qu et al., 2008**; **May-Zhang et al., 2021**; **Morarach et al., 2021**). We found that all *Nmu+* neurons co-express *Calcb* in both jejunum and distal colon (**Figure 1I–K**). In contrast, only about half of *Calcb+* neurons in the jejunum and two-thirds in the distal colon co-express *Nmu* (**Figure 1L**).

We next assessed co-expression of *Cdh6* with *Calcb* and *Nmu*. In the jejunum, we found that nearly all *Nmu+* neurons and *Calcb+* neurons express *Cdh6* (**Figure 1P and U**), though only about three-quarters of all *Cdh6+* neurons express *Calcb* and only about half express *Nmu* (**Figure 1O and T**). In contrast, in the distal colon, while *Cdh6* is only expressed in about two-thirds of all *Calcb+* neurons (**Figure 1U**), nearly all *Cdh6+* neurons express *Nmu* and *Calcb* (**Figure 1O and T**). Taken together, our data show that in the myenteric plexus, *Cdh6* is expressed exclusively in *Calcb+/Nmu+* IPANs in the mouse distal colon (**Figure 1W and X**).

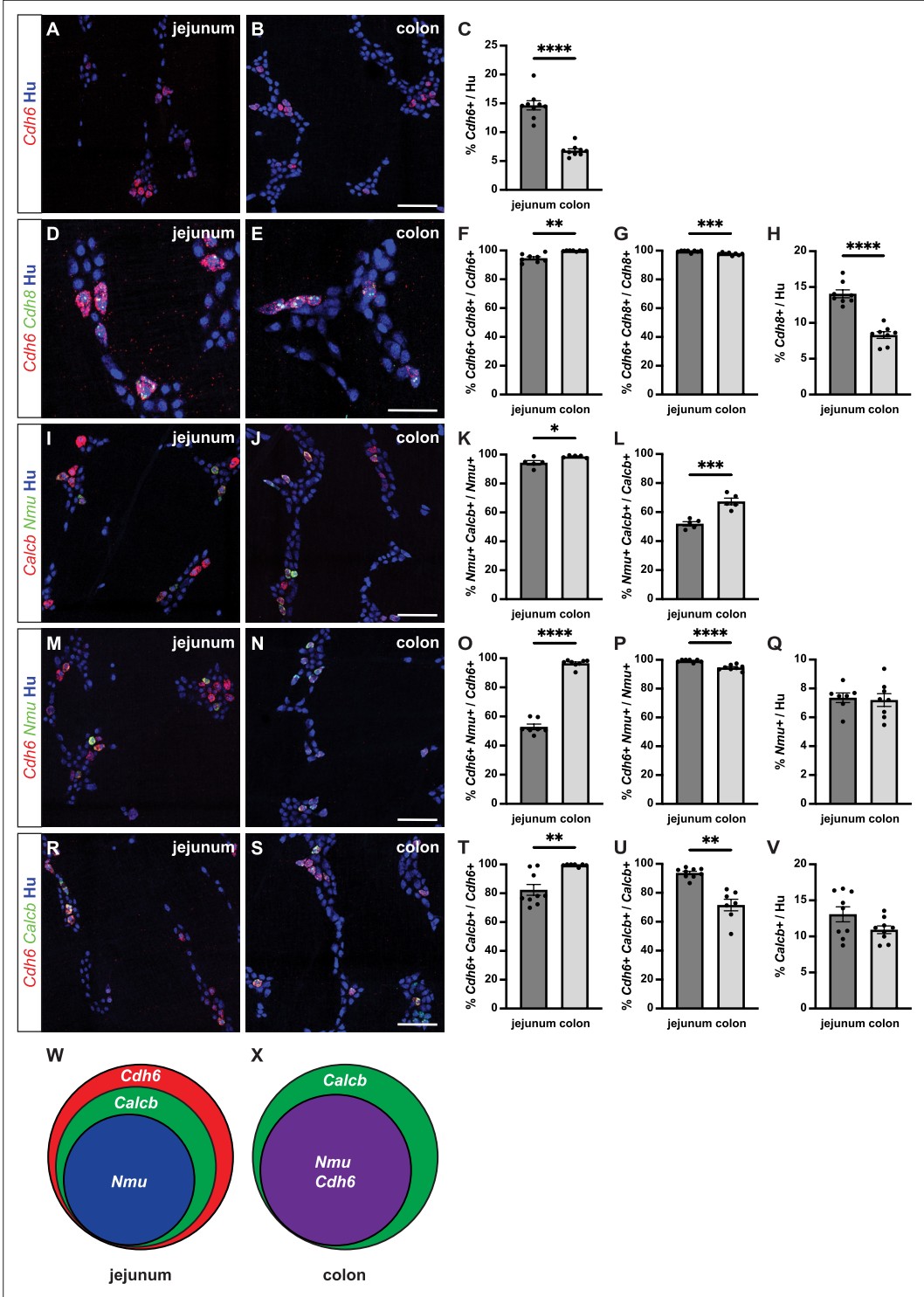

**Figure 1.** *Cdh6* expression overlaps with intrinsic primary afferent neuron (IPAN) markers *Calcb* and *Nmu*. (**A,
B**) Representative images of jejunum (**A**) and distal colon (**B**) myenteric plexus labeled with HuC/D (IHC) (blue)
and *Cdh6* (RNA) (red). (**C**) Proportion of total HuC/D neurons positive for *Cdh6* (jejunum, n=9; distal colon, n=9).
(**D, E**) As in (**A, B**) for HuC/D (IHC) (blue), *Cdh6* (RNA) (red), and *Cdh8* (RNA) (green). (**F**) Proportion of *Cdh6+*
neurons positive for *Cdh8* (jejunum, n=8; distal colon, n=8). (**G**) Proportion of *Cdh8+* neurons positive for *Cdh6*
(jejunum, n=8; distal colon, n=8). (**H**) Proportion of total HuC/D neurons positive for *Cdh8* (jejunum, n=8; distal
colon, n=8). (**I, J**) As in (**A, B**) for HuC/D (IHC) (blue), *Calcb* (RNA) (red), and *Nmu* (RNA) (green). (**K**) Proportion
of *Nmu+* neurons positive for *Calcb* (jejunum, n=5; distal colon, n=5). (**L**) Proportion of *Calcb+* neurons positive

*Figure 1 continued on next page*

*Figure 1 continued*

for *Nmu* (jejunum, n=5; distal colon, n=5). (**M, N**) As in (**A, B**) for HuC/D (IHC) (blue), *Cdh6* (RNA) (red), and *Nmu* (RNA) (green). (**O**) Proportion of *Cdh6+* neurons positive for *Nmu* (jejunum, n=7; distal colon, n=8). (**P**) Proportion of *Nmu+* neurons positive for *Cdh6* (jejunum, n=7; distal colon, n=8). (**Q**) Proportion of total HuC/D neurons positive for *Nmu* (jejunum, n=7; distal colon, n=8). (**R, S**) As in (**A, B**) for HuC/D (IHC) (blue), *Cdh6* (RNA) (red), and *Calcb* (RNA) (green). (**T**) Proportion of *Cdh6+* neurons positive for *Calcb* (jejunum, n=9; distal colon, n=7). (**U**) Proportion of *Calcb+* neurons positive for *Cdh6* (jejunum, n=9; distal colon, n=7). (**V**) Proportion of total HuC/D neurons positive for *Calcb* (jejunum, n=9; distal colon, n=9). (**W, X**) Schematic of marker overlap in jejunum (**W**) and distal colon (**X**). Scale bar represents 100 μm for (**A, B, I, J, M, N, R, S**), 50 μm for (**D, E**). All charts (mean ± SEM). *p<0.05; **p<0.01; ***p<0.001; ****p<0.0001.

## Mouse colonic IPANs display AH-type electrophysiology and $I_H$ current

We next assessed the electrophysiological properties of *Cdh6+* IPANs. We focused our analysis on the colon, and for ease of neuron tracing, took advantage of a genetic strategy to sparsely label *Cdh6+* neurons. Previous studies of Hb9:GFP transgenic mice have shown that due to the inserted transgene's proximity to *Cdh6*, *Cdh6+* neurons can express eGFP (*Laboulaye et al., 2018*). Hb9:GFP+ neurons were rare and projected extensively throughout the myenteric plexus (*Figure 2A–C*). In situ hybridization confirmed *eGFP* expression was limited to a small fraction (3.5 ± 0.8%) of *Cdh6+* colonic neurons (*Figure 2D and F–H*), and all *eGFP+* neurons expressed *Cdh6* (*Figure 2E*).

We developed a protocol to perform whole-cell patch-clamp recordings (*Osorio and Delmas, 2011*) in Hb9:GFP+ colonic neurons in the distal colon. Membrane capacitance reflecting overall size of these cells was 32±8.7 pF (*Figure 3D*); their resting membrane potential (RMP) was –49.4±2.9 mV (*Figure 3C*). The input resistance ($R_{in}$) was 393±54.7 MΩ (*Figure 3E*) as computed from the slope of the voltage-current (V-I) relationship. All patched neurons had large-amplitude AP (72±2.5 mV, *Figure 3F and E*) with threshold of –26.4±0.9 mV (*Figure 3L*) and a half-width of 1.2±0.1 ms (*Figure 3H*) elicited

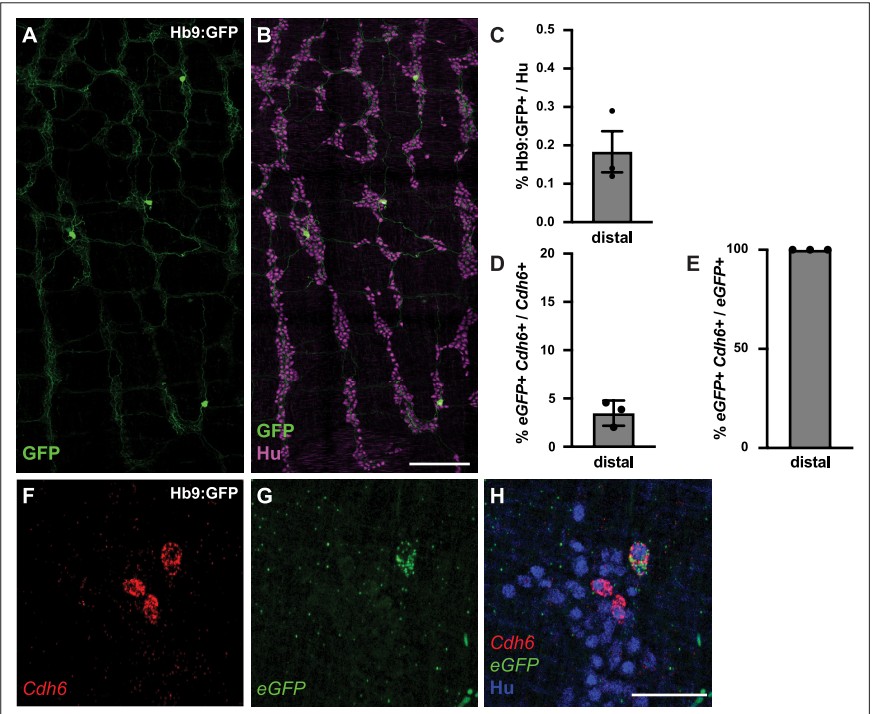

**Figure 2.** Hb9:GFP+ is expressed in a small proportion of *Cdh6+* colon myenteric neurons. (**A, B**) Representative images of Hb9:GFP+ distal colon myenteric plexus labeled with HuC/D (IHC) (magenta) and GFP (green). (**C**) Proportion of total distal colon HuC/D neurons positive for GFP (n=3). (**D**) Proportion of distal colon *Cdh6+* neurons positive for *eGFP* (n=3). (**E**) Proportion of distal colon *eGFP+* neurons positive for *Cdh6* (n=3). (**F–H**) Representative images of Hb9:GFP+ distal colon myenteric plexus labeled with HuC/D (IHC) (blue), *Cdh6* (RNA) (red), and e*GFP* (RNA) (green). Scale bar represents 200 μm for (**A, B**), 50 μm for (**F-H**). All charts (mean ± SEM).

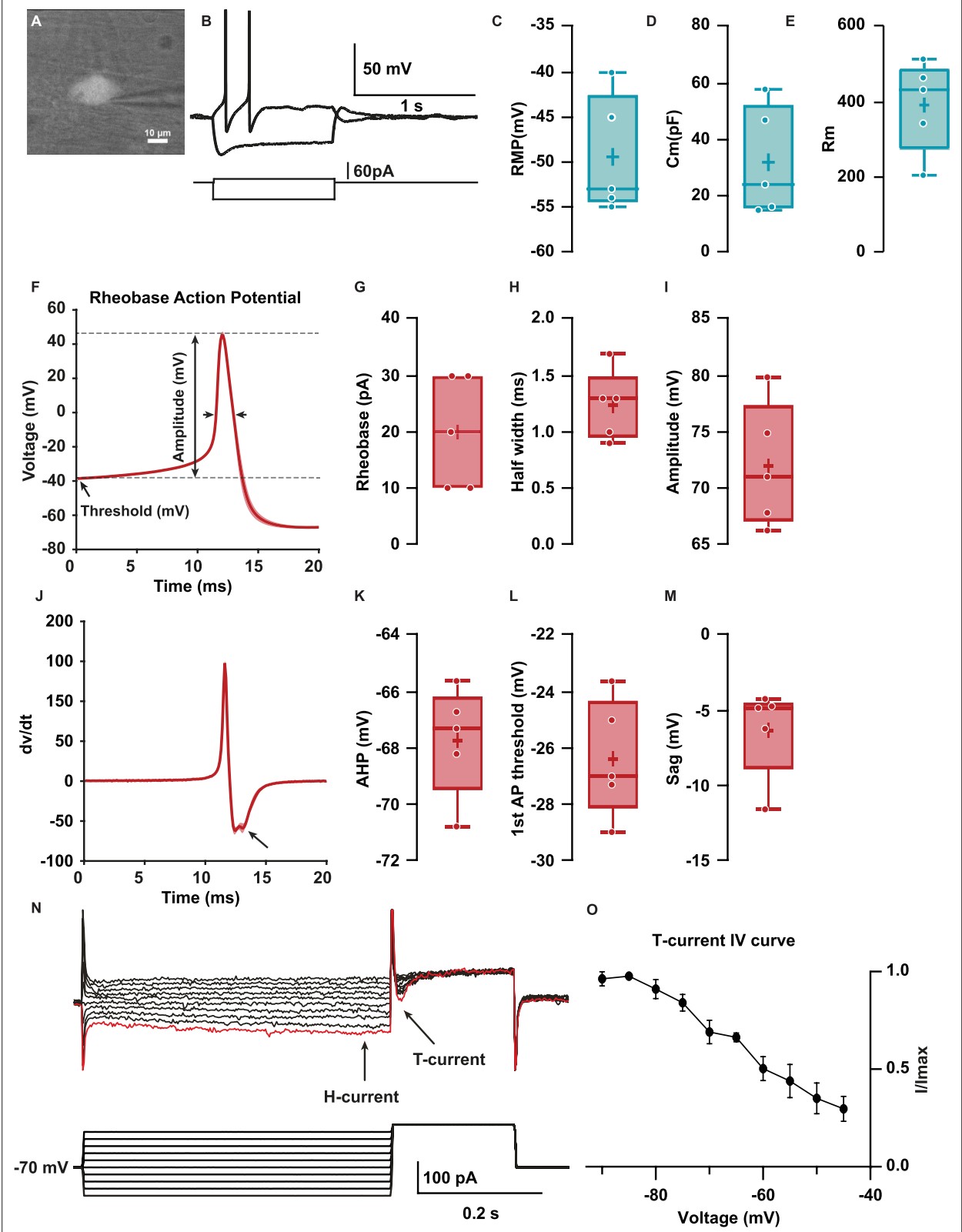

**Figure 3.** Hb9:GFP+ distal colon neurons have afterhyperpolarizing (AH) electrophysiological characteristics. (**A**) IR videomicroscopy image of an Hb9:GFP distal colon neuron that presented a large soma located in a ganglion (scale bar, 10 μm). (**B**) Current-clamp recordings of the same neuron in (**A**) obtained in response to application of current pulse (bottom traces) of –50 pA and +10 pA. Note the presence of a sag and a post-hyperpolarization rebound depolarization. (**C–E**) Box-and-whisker plots of cellular properties of recorded neurons. (**C**) Resting membrane potential (RMP), (**D**) capacitance

**Figure 3 continued**

(Cm), and (**E**) membrane resistance (Rm) (N=5). (**F**) Averaged traces of the first spike (rheobase action potential) after a depolarization step of 1 s. (**J**) Averaged derivative traces of the first spike (rheobase action potential). An inflection on the repolarizing phase is observed in the first derivative (arrow). (**G–I, K–N**) Box-and-whisker plots of electrophysiological properties of recorded neurons; rheobase action potential (AP, **G–I**) (**G**) current threshold, (**H**) half-width, (**I**) amplitude, (**K**) afterhyperpolarization (AHP), and (**L**) threshold. (**M, N**) Non-AP properties sag (mV) and rebound (mV). (**O**) H and T currents in recorded neurons. Top: example of currents obtained from voltage protocol. Bottom: 500 ms hyperpolarizations ranging from –90 to –45 for 500 ms followed by depolarizing to –40 mV. Hyperpolarizations evoked slowly activating inward current (H-current, arrow), followed by a transient inward current upon post-conditioning step to –30 mV (T-current, arrow). Largest T and H currents were obtained with the most hyperpolarized potentials (red trace). (**P**) Normalized peak $I_T$ plotted versus holding potential to obtain the $I/I_{max}$ curve (N=3/3). Scale bar represents 10 μm for (**A**).

at rheobase (20±4.5 pA, **Figure 3G**), each followed by an afterhyperpolarization (AHP = –67.7 ± 0.9 mV, **Figure 3K**). In addition, the first derivative of membrane voltage during the AP (dV/dt, **Figure 3J**) exhibited an inflection during the repolarization phase, suggesting the presence of fast and slow AP repolarization mechanisms (**Figure 3F**). Patched Hb9:GFP+ neurons generally could not sustain repetitive APs in response to long depolarizing current pulses. All patched neurons responded to hyperpolarizing current pulses with a time-dependent membrane potential sag (–6.3±1.2 mV, **Figure 3B and M**) and a rebound depolarization following the hyperpolarization (**Figure 3B and N**).

Consistent with these findings of sag and rebound, voltage clamp step hyperpolarizations revealed $I_H$ (hyperpolarization-activated current, **Figure 3N**) and, upon repolarization, $I_T$ (transient inward presumed $Ca^{2+}$ current) (**Huguenard, 1996**) in terms of its kinetics (**Figure 3O**) and steady-state inactivation (**Figure 3O and P**). Taken together, these results show that Hb9:GFP+/*Cdh6+* distal colonic neurons have AP afterhyperpolarizing (AH)-type electrophysiology typical of IPANs, including rhythm generating currents $I_H$ and $I_T$ (**Xiao et al., 2004**; **Nurgali et al., 2007**; **Mao et al., 2006**).

## Colonic IPANs have Dogiel type II morphology and abundant projections throughout the myenteric plexus

To visualize the morphology and projections of individual patched Hb9:GFP+ neurons, we included biocytin in the internal solution for postfixation single-cell tracing (**Figure 4**). Patched Hb9:GFP+ neurons displayed Dogiel type II morphology (**Furness et al., 2004**), with large smooth cell somas and multiple branching neurites. Projections were mainly circumferential and extensively branched within myenteric ganglia. Thus, Hb9:GFP+ neurons display morphological features characteristic of IPANs (**Nurgali et al., 2004**; **Furness et al., 2004**).

To further visualize the full extent of IPAN circuitry in the myenteric plexus, we intercrossed Cdh6$^{CreER}$ (**Kay et al., 2011**) and ROSA26$^{LSL-tdTomato}$ (Ai14) (**Madisen et al., 2010**) mice and induced Cre expression at 5–8 weeks of age (**Figure 5A and B**). We confirmed tdTomato+ labeling in myenteric *Cdh6+* neurons (**Figure 5E and F**), representing about 5% of the total neuronal population (**Figure 5G**). tdTomato+ neurons had large cell somas (major axis, 27.8±0.7 μm; minor axis, 15.9±0.4 μm) (**Figure 5D**). All ganglia of the myenteric plexus were densely innervated by tdTomato+ fibers (**Figure 5A and B**), which also projected into the circular muscle. We noted additional tdTomato+ labeling of some putative longitudinal and circular muscle cells (**Figure 5A and B**). Taken together, our data reveal an IPAN array that spans the entire motility circuitry of the colonic myenteric plexus.

## Optogenetic activation of distal colon IPANs evokes CMCs

In thalamocortical relay neurons, $I_H$ contributes to intrinsic slow rhythmic burst firing at 1–2 Hz (**McCormick and Pape, 1990**), but the function of $I_H$ in colonic IPANs is not known. During CMCs, large regions of the ENS oscillate in synchrony at 1–2 Hz to generate traveling contractions along the colon (**Spencer et al., 2021**). Recent calcium imaging studies have shown that IPANs participate, along with all other subtypes of enteric neurons, in this synchronized oscillatory firing (**Hibberd et al., 2018b**). Furthermore, our electrophysiological studies confirm the presence of $I_H$ in mouse colonic IPANs (**Figure 3N**). However, the role of IPANs in spontaneous CMCs is not well understood.

To interrogate the functional role of *Cdh6+* IPANs in CMCs, we performed ex vivo colonic contraction force recordings in conjunction with optogenetic activation. To express ChR2-eYFP in *Cdh6+* cells, we intercrossed Cdh6$^{CreER}$ and ROSA26$^{LSL-ChR2-eYFP}$ (Ai32) (**Madisen et al., 2012**) mice and induced Cre expression at 5–8 weeks of age (**Figure 6A–C**). In Cdh6$^{CreER}$+;ChR2-eYFP+ colon preparations, we observed spontaneous CMCs at regular intervals of about 3–5 min. Blue light stimulation of *Cdh6+*

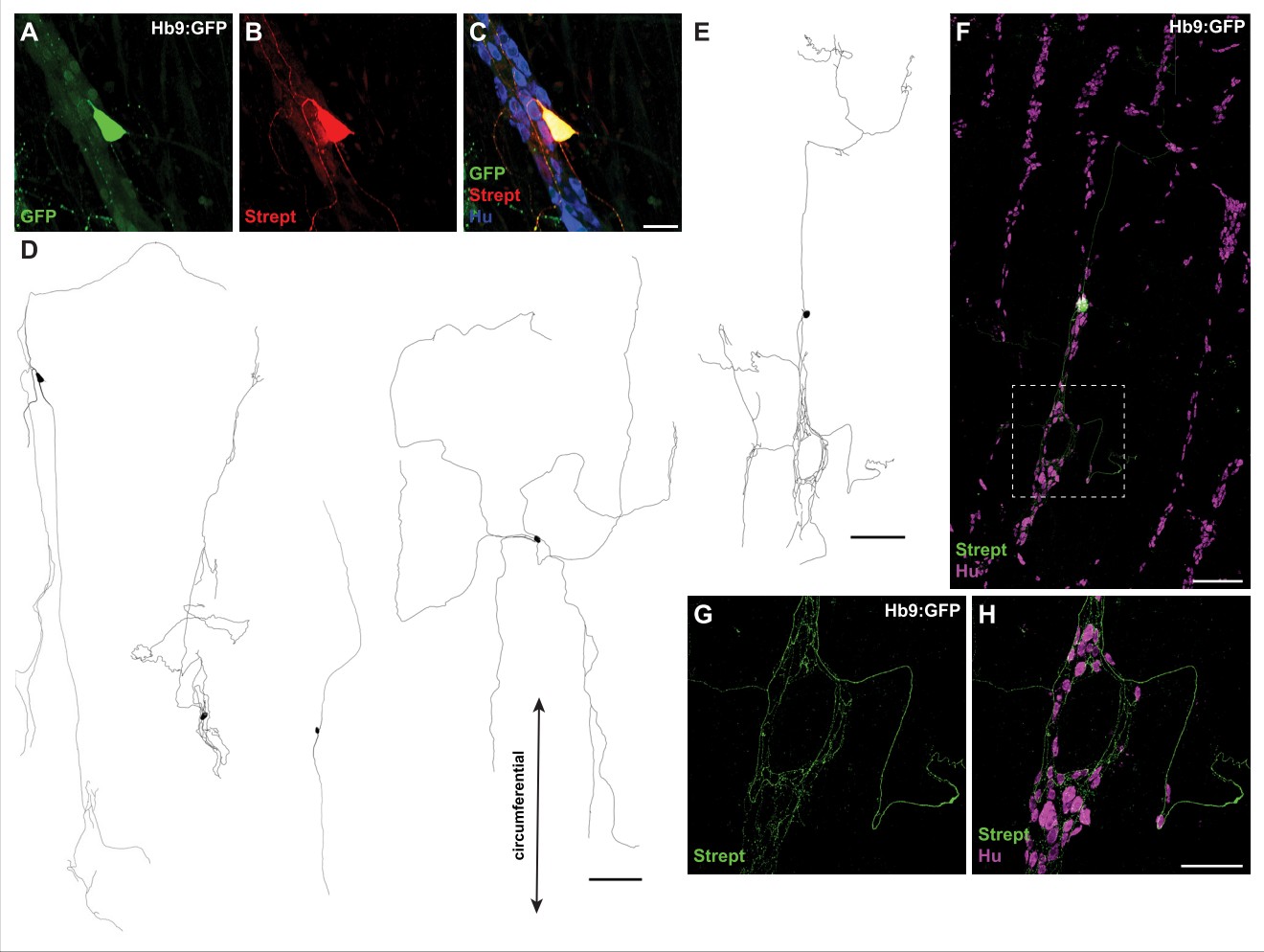

**Figure 4.** Hb9:GFP+ distal colon neurons have circumferential branching projections. (**A–C**) Representative images of Hb9:GFP+ distal colon myenteric plexus labeled with HuC/D (blue), streptavidin (red), and GFP (green). (**D, E**) Tracings of Hb9:GFP+ distal colon neurons filled with biocytin during whole-cell patch-clamp recording. (**F**) Image of patched and filled Hb9:GFP+ distal colon neuron traced in (**E**). (**G, H**) Inset of (**F**). Scale bar represents 40 μm for (**A–C**), 200 μm for (**D–F**), 100 μm for (**G, H**).

IPANs in distal colon 60–90 s after a spontaneous CMC ('control' CMC) resulted in an evoked, premature CMC that began during stimulation (N=17 stimulations; n=5 mice) (*Figure 6D and E*). Evoked CMCs traveled retrogradely from the distal to the proximal colon. They were similar to spontaneous CMCs in peak amplitude, area under the curve, and duration, though the contractile force (peak amplitude and AUC) was slightly weaker in the proximal colon (*Figure 6F–H*). Blue light stimulation in proximal or mid colon failed to generate CMCs (n=5/5, data not shown). In comparison, stimulation in control Cdh6$^{CreER}$-negative;ChR2-eYFP+ colons failed to evoke any CMCs (N=28 stimulations; n=7 mice) (*Figure 6—figure supplement 1*).

CMCs have previously been shown to depend on nicotinic cholinergic transmission (*Hibberd et al., 2018b*). We performed optogenetic stimulation in the presence of hexamethonium, a blocker of nicotinic cholinergic transmission. Spontaneous CMCs were abolished in hexamethonium, and CMCs could not be evoked by optogenetic stimulation (*Figure 6I*). We conclude that activation of *Cdh6+* distal colon IPANs evokes retrograde-traveling but otherwise characteristic and hexamethonium-sensitive CMCs.

## Blockade of $I_H$ current in colonic IPANs disrupts CMC production

To determine whether $I_H$ in IPANs may contribute to oscillatory firing driving CMCs, we measured colonic contraction force on a tethered pellet in the presence of $I_H$ blockers ZD7288 or CsCl (*Harris*

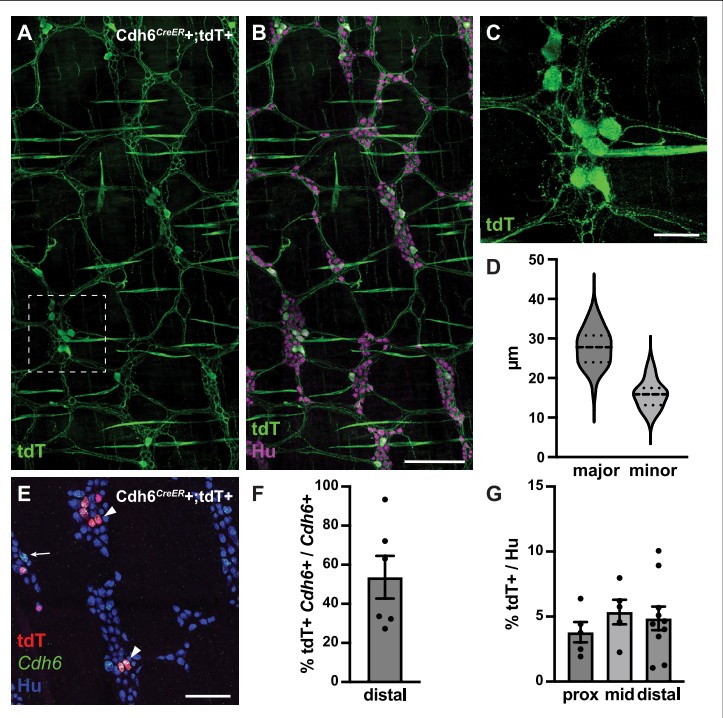

**Figure 5.** Cdh6$^{CreER}$+/tdTomato+ neurons have Dogiel type II morphology. (**A, B**) Representative images of Cdh6$^{CreER}$+;tdTomato+ distal colon myenteric plexus labeled with HuC/D (IHC) (magenta) and tdTomato (IHC) (green). (**C**) Inset of (**A**). (**D**) Dimensions of tdTomato+ neurons (major and minor axes) (N=73; n=3). (**E**) Representative image of Cdh6$^{CreER}$+;tdTomato+ distal colon myenteric plexus labeled with HuC/D (IHC) (blue), tdTomato (IHC) (red), and *Cdh6* (RNA) (green). Arrowheads indicate *Cdh6*+/tdTomato+ cells; arrow, *Cdh6*+/tdTomato-negative cell. (**F**) Proportion of *Cdh6*+ distal colon neurons positive for tdTomato (n=6). (**G**) Proportion of total HuC/D neurons positive for tdTomato (proximal colon, n=5; mid colon, n=5; distal colon, n=10). Scale bar represents 100 μm for all images. All charts (mean ± SEM).

*and Constanti, 1995*; *Galligan et al., 1990*). ZD7288 blocks $I_H$ in all IPANs, including *Cdh6*+ IPANs. Spontaneous CMCs were recorded in all preparations of both proximal and distal colon prior to drug application. Addition of 10 μM ZD7288 to the recording chamber abolished spontaneous CMCs and washout of ZD7288 recovered spontaneous CMC activity (*Figure 6J*, *Figure 6—figure supplement 2*). Addition of 2 mM CsCl also impaired or altered spontaneous rhythmic production of typical CMCs (*Figure 6K*, *Figure 6—figure supplement 3*); rhythmic contractions increased in frequency (proximal colon, n=5/6; distal colon, 6/6), decreased in amplitude (proximal colon, n=5/6; distal colon, n=4/6, p=0.0606), or in some cases even included significant retrograde force components (proximal colon, n=2/6; distal colon, n=2/6). We conclude that pharmacologic blockade of $I_H$ in IPANs impairs the production of CMCs in the mouse colon.

## Discussion

Our study shows that in the myenteric plexus, *Cdh6* is expressed exclusively in *Calcb*+/*Nmu*+ IPANs located in the mouse distal colon, while in the small intestine, *Cdh6* is also expressed in some *Calcb*+/*Nmu*- and *Calcb*-/*Nmu*- neurons. We confirm the IPAN identity of *Cdh6*+ distal colonic neurons by electrophysiological recordings revealing AH-type signature, and single neuron tracings showing Dogiel type II morphology. Finally, we demonstrate that activation of IPANs in the distal colon evokes retrograde CMCs, while pharmacologic blockade of $I_H$, a rhythmicity-associated current we show also present in mouse colonic IPANs, disrupts spontaneous CMC generation.

Together with *Cdh8*, which we show to be co-expressed with *Cdh6*, our study validates two new adhesion molecules specific to IPANs in the distal colon. Notably, these markers show a different expression pattern than another cadherin, *Cdh9*, which is exclusively expressed in the small intestine

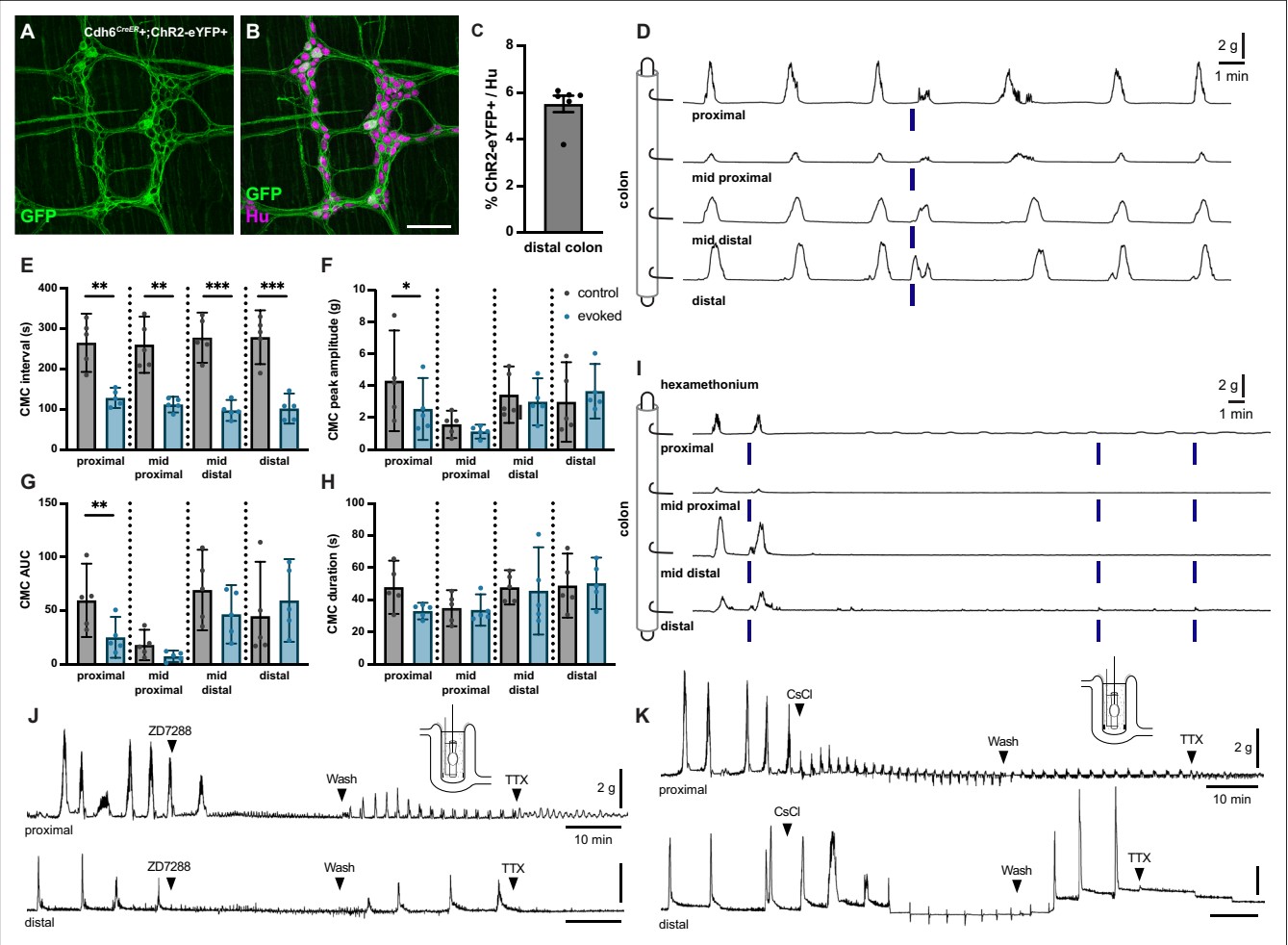

**Figure 6.** Optogenetic stimulation of distal colonic *Cdh6+* neurons evokes colonic motor complexes (CMCs), while pharmacologic blockade of $I_H$ abolishes spontaneous CMCs. (**A, B**) Representative images of Cdh6$^{CreER}$+;ChR2-eYFP+ distal colon myenteric plexus labeled with HuC/D (magenta) and GFP (green). (**C**) Proportion of total distal colon HuC/D neurons positive for ChR2-eYFP (n=6). (**D**) Representative force traces. Blue bars indicate timing of light stimulation. LEDs placed distal to distal hook. (**E**) CMC intervals recorded from force traces. Evoked (blue) intervals represent the time from the prior spontaneous CMC before stimulation to the evoked CMC following stimulation. Control (gray) intervals represent the time between the spontaneous CMC prior to stimulation and the previous spontaneous CMC (n=5). Paired t test, one-tailed. (**F**) CMC peak amplitude recorded from force traces. Evoked (blue) indicates the evoked CMC following stimulation. Gray (control) indicates the spontaneous CMC prior to stimulation (n=5). Paired t test, two-tailed. (**G**) CMC AUC (area under the curve). Evoked (blue) and control (gray) as in (**F**) (n=5). Paired t test, two-tailed. (**H**) CMC duration. Evoked (blue) and control (gray) as in (**F**) (n=5). Paired t test, two-tailed. (**I**) Representative force traces. Hex indicates addition of 300 µM hexamethonium. Blue bars indicate timing of light stimulation. LEDs placed distal to distal hook (n=5/5). (**J**) Representative force traces on tethered pellets. First arrowhead indicates addition of 10 µM ZD7288. Second arrowhead indicates washout in Krebs. Third arrowhead indicates addition of 1 µM TTX. ZD7288 abolished CMCs in both proximal and distal colon (n=6/6, p=0.0022, Fisher's exact test). Washout in Krebs restored CMCs in both proximal and distal colon (n=6/6, p=0.0022, Fisher's exact test). (**K**) As in (**J**). First arrowhead indicates addition of 2 mM CsCl. Second arrowhead indicates washout in Krebs. Third arrowhead indicates addition of 1 µM TTX. Typical CMC production was impaired or altered by CsCl (proximal colon, n=5/6, p=0.0152; distal colon, n=6/6, p=0.0022, Fisher's exact test): increased frequency (proximal colon, n=5/6, p=0.0152; distal colon, n=6/6, p=0.0022, Fisher's exact test), decreased in amplitude (proximal colon, n=5/6, p=0.0152; distal colon, n=4/6, p=0.0606, Fisher's exact test); retrograde force (proximal colon, n=2/6; distal colon, n=2/6). Scale bar represents 100 µm for (**A, B**). *p<0.05; **p<0.01; ***p<0.001.

The online version of this article includes the following figure supplement(s) for figure 6:

**Figure supplement 1.** Optogenetic stimulation control.

**Figure supplement 2.** Pharmacologic blockade of $I_H$ with ZD7288 abolishes spontaneous colonic motor complexes (CMCs).

**Figure supplement 3.** Pharmacologic blockade of $I_H$ with CsCl impairs generation of colonic motor complexes (CMCs).

in mouse, not in the colon (**May-Zhang et al., 2021**). *Cdh9* is expressed in a subset of IPANs not expressing either *Calcb* or *Nmu* (**Qu et al., 2008**; **May-Zhang et al., 2021**; **Morarach et al., 2021**). It is possible that in the small intestine *Cdh6* and *Cdh9* mark some of the same neurons. However, RNA-Seq data from two separate studies suggest this is unlikely (**Drokhlyansky et al., 2020**; **Morarach et al., 2021**). We conclude that *Cdh6/Cdh8* IPANs are a separate population from *Cdh9* IPANs. Finally, our study was limited to the myenteric plexus, containing motility circuitry. IPANs are also present in the submucosal plexus, and it would be interesting to investigate *Cdh6* expression and neuronal subtype in the SMP.

Though IPANs are positioned to initiate motility by activating other subtypes in enteric circuitry, their role in spontaneous CMCs has been debated, as CMCs can occur in the absence of luminal contents, without any apparent stimulus for IPANs to sense, or upon stimulation of nitrergic populations (**Nurgali et al., 2004**; **Koh et al., 2022**). Only recently has evidence emerged to suggest that IPANs participate in oscillatory rhythmic firing of the ENS during CMCs, and that activation of neuronal subtypes expressing calretinin, including IPANs, together can evoke CMCs (**Hibberd et al., 2018b**; **Hibberd et al., 2018a**). Our results demonstrate that excitement of IPANs alone in the distal colon is capable of producing retrograde CMCs. The exact mechanism causing CMC generation at an established and controlled frequency of once every few minutes, without evident stimulus, remains to be determined.

A major observation was that optogenetic stimulation of *Cdh6+* neurons readily evoked CMCs from the distal colon, but never from the proximal colon. Failure of proximal colonic stimulation to evoke CMCs may reflect inhibition by IPAN recruitment of descending pathways; or conversely, high efficacy of stimulation from distal colon may reflect bias toward activation of ascending excitatory cholinergic pathways. Our classification of *Cdh6+* neurons as IPANs in the distal colon may not extend to the proximal colon; thus, we cannot extrapolate that *Cdh6+* optogenetic activation in the colon is restricted to IPANs as in the distal colon. In addition, our stimulus paradigm in the proximal colon may not have activated enough neurons based on light density and illumination. In previous studies, optogenetic stimulation of calretinin-expressing neurons or choline acetyl transferase-expressing neurons (both of which include IPANs, interneurons, and motor neurons) elicited anterograde CMCs regardless of stimulus location, including proximal colon (**Hibberd et al., 2018a**; **Efimov et al., 2024**). Proximal colon stimulation of nitric oxide synthase-expressing neurons also elicited CMCs (**Koh et al., 2022**). These studies, as noted, each activated multiple neuronal classes employing the same neurotransmitter. How broad and nonspecific activation of such disparate neuron classes readily evokes CMCs remains unclear.

Spontaneous and evoked CMCs were abolished in hexamethonium, confirming that CMC synchronous firing is dependent on nicotinic cholinergic transmission (**Hibberd et al., 2018b**). This reinforces that nicotinic cholinergic transmission is required for greater activation and synchrony of the entire ENS motility network to generate CMCs.

Through our electrophysiological investigation of IPANs in the distal colon using voltage clamp, our work reveals the presence of two voltage-gated ion conductances and their underlying currents, $I_T$ and $I_H$. Slow AHP, $I_H$, and $I_T$ have been previously identified as distinguishing characteristics of IPANs in rat and guinea pig (**Xiao et al., 2004**; **Nurgali et al., 2007**; **Mao et al., 2006**). Although prior studies noted $I_H$ and proposed $I_T$ in guinea pig Dogiel type II neurons (**Nurgali et al., 2007**), they have not previously been reported in studies of intact mouse distal colon myenteric plexus due to the conventional reliance on sharp electrode recordings in which voltage clamp is not possible (**Li et al., 2022**). The presence of $I_H$ and $I_T$ in thalamocortical relay neurons and other cell types supports intrinsic rhythmicity (**McCormick and Pape, 1990**; **Pape and McCormick, 1989**). It is possible that these two currents may similarly promote autonomous rhythmic activity in colonic IPANs.

$I_H$ is conducted through hyperpolarization-activated cyclic nucleotide-gated (HCN) channels (**Benarroch, 2013**). HCN channel family members HCN1 and HCN2 have been shown to be present in mouse distal colonic Dogiel type II neurons (**Xiao et al., 2004**), and RNA-Seq ENS screens similarly indicate their expression in IPANs (**Drokhlyansky et al., 2020**; **Morarach et al., 2021**). Knockout of HCN2 in mouse leads to a severe growth restriction phenotype due to malnutrition and GI dysmotility (**Fisher et al., 2018**). Here, we demonstrate that pharmacologic blockade of $I_H$, which we show to be present in IPANs, with two distinct HCN channel blockers ZD7288 and CsCl abolishes CMCs, an otherwise persistent and ongoing pattern of motor activity in the mouse colon. We speculate that $I_H$

in colonic IPANs, as in thalamocortical neurons, plays a role in promoting either rhythmic oscillatory single neuron activity or network burst firing, or both (*McCormick and Pape, 1990*; *Pape and McCormick, 1989*). Blocking $I_H$ may impair individual IPANs' ability to fire rhythmically, or the ability of IPANs to synchronize into a network burst firing mode. Failure of IPANs to fully activate and synchronize could prevent generation of both (*Fung and Vanden Berghe, 2020*) synchronized rhythmic myenteric network activation of motility circuits, and (*Nurgali et al., 2004*) the resulting synchronized contractions that sum to much larger contractile forces during CMCs.

Type II cadherins are most commonly homophilic synaptic cell adhesion molecules (*Yamagata et al., 2018*; *Basu et al., 2015*). *Cdh6* can also form heterodimers with *Cdh7*, *Cdh10*, and *Cdh14* (*Shimoyama et al., 2000*). However, *Cdh10* and *Cdh14* are very lowly expressed in the colon by single-cell RNA sequencing (*Drokhlyansky et al., 2020*), and we performed RNAscope for *Cdh7* and did not observe any expression (data not shown). Restricted expression of two type II cadherins, *Cdh6* and *Cdh8*, to mouse colonic IPANs raises the possibility of these cadherins supporting IPAN-IPAN synaptic connections. While broadly speaking, IPANs are not known to receive synaptic input and in fact have been characterized electrophysiologically by their lack thereof (*Hirst et al., 1974*), some work has in fact suggested that AH-AH neuron interconnected pairs may exist (*Kunze et al., 1993*). Immunohistochemical and electron microscopy investigation of synapses on enteric neurons further showed that calbindin-positive neurons, presumed IPANs, do receive synapses, though fewer than non-calbindin neurons, and some of those synapses were also calbindin-reactive (*Pompolo and Furness, 1988*). These observations informed a proposed 'IPAN driver circuit' theory, in which IPANs form an interconnected network of positive feedback to synchronize and amplify sensory signaling and thus activate large swaths of enteric circuitry (*Wood, 2012*). In contrast, more recently, activation of large regions of the ENS has been suggested to be driven by interneuronal networks (*Barth et al., 2022*). It is important to note that our study was unsuccessful in localizing Cdh6 protein via immunohistochemistry, to confirm protein expression in neurons or in Cdh6-cre-tdT+ muscle cells or visualize synaptic connections between neurons or connections to other cell types. Further investigations are necessary to determine whether synaptic adhesion molecules, such as *Cdh6* and *Cdh8*, may in fact support IPAN-IPAN synapses underlying an 'IPAN driver circuit'.

## Materials and methods

**Key resources table**

| Reagent type (species) or resource | Designation | Source or reference | Identifiers | Additional information |
|---|---|---|---|---|
| Strain, strain background (*Mus musculus*) | C57BL/6J | Jackson Laboratory | #000664 | |
| Strain, strain background (*M. musculus*) | Hb9:GFP | Jackson Laboratory | #005029 | |
| Strain, strain background (*M. musculus*) | Ai14 | Jackson Laboratory | #007908 | |
| Strain, strain background (*M. musculus*) | Ai32 (ChR2-eYFP) | Jackson Laboratory | #024109 | |
| Strain, strain background (*M. musculus*) | Cdh6$^{CreER}$ | Xin Duan, UCSF | | |
| Antibody | Human anti-HuC/D | Vanda Lennon, Mayo Clinic | | IF(1:75,000) |
| Antibody | Sheep anti-GFP (polyclonal) | Biogenesis | Cat# 4745-1051, RRID:AB_619712 | IF(1:1000) |
| Antibody | Rabbit anti-RFP (polyclonal) | Rockland | Cat# 600-401-379, RRID:AB_2209751 | IF(1:1000) |
| Antibody | Rabbit anti-PGP9.5 (polyclonal) | Abcam | #ab15503, RRID:AB_301912 | IF(1:4000) |
| Antibody | Donkey anti-human Alexa Fluor (AF)-647 (polyclonal) | Jackson ImmunoResearch | #709-605-098, RRID:AB_2340577 | IF(1:500) |

*Continued on next page*

*Continued*

| Reagent type (species) or resource | Designation | Source or reference | Identifiers | Additional information |
|---|---|---|---|---|
| Antibody | Donkey anti-sheep AF-488 (polyclonal) | Invitrogen | #A11015, RRID:AB_141362 | IF(1:1000) |
| Antibody | Donkey anti-rabbit AF-488 (polyclonal) | Invitrogen | #A21206, RRID:AB_2535792 | IF(1:1000) |
| Peptide, recombinant protein | Streptavidin AF-546 | Invitrogen | #S11225 | IF(1:500) |
| Sequence-based reagent | RNAscope probe Mm-Cdh6 | Advanced Cell Diagnostics | Cat #519541 | |
| Sequence-based reagent | RNAscope probe Mm-Cdh8 | Advanced Cell Diagnostics | Cat #485461 | |
| Sequence-based reagent | RNAscope probe Mm-Nmu | Advanced Cell Diagnostics | Cat #446831 | |
| Sequence-based reagent | RNAscope probe Mm-Calcb | Advanced Cell Diagnostics | Cat #425511 | |
| Sequence-based reagent | RNAscope probe Mm-eGFP | Advanced Cell Diagnostics | Cat #400281 | |
| Commercial assay or kit | RNAscope Multiplex Fluorescent V2 Assay kit with RNA-Protein Co-detection Ancillary Kit | Advanced Cell Diagnostics | Cat #323100 | |
| Chemical compound, drug | Protease XIV | Sigma | #P5417 | |
| Chemical compound, drug | Collagenase | Worthington | #CLS-4 | |
| Chemical compound, drug | Dispase | Sigma | #D4693 | |
| Chemical compound, drug | ZD7288 | Sigma-Aldrich | #73777 | |
| Chemical compound, drug | Cesium chloride | Sigma-Aldrich | #C4036 | |
| Chemical compound, drug | Tetrodotoxin citrate | Alomone Labs | #T-550 | |
| Software, algorithm | Imaris Filament Tracer | Bitplane, Oxford Instruments | | |
| Software, algorithm | LabChart 7, 8 | AD Instruments | | |
| Software, algorithm | Prism 9 | GraphPad | | |

## Mice

All procedures conformed to the National Institutes of Health Guidelines for the Care and Use of Laboratory Animals and were approved by the Stanford University Administrative Panel on Laboratory Animal Care. Mice were group housed up to a maximum of five adults per cage. Food and water were provided ad libitum and mice were maintained on a 12:12 LD cycle. Male and female mice were used in all experiments.

Wild-type C57BL/6J mice (#000664), Hb9:GFP mice (#005029), Ai14 (#007908), and Ai32 (#024109) (*Madisen et al., 2010*) mice were obtained from the Jackson Laboratory and from the Animal Resource Center (ARC) in Western Australia, with JAX heritage. Cdh6$^{CreER}$ mice (#029428) (*Kay et al., 2011*) were provided by Xin Duan (UCSF). Cdh6$^{CreER}$ mice were crossed to Ai14 mice and Ai32 mice to generate mice heterozygous for each allele, termed Cdh6$^{CreER}$;Ai14 and Cdh6$^{CreER}$;Ai32. Tamoxifen (20 mg/mL in corn oil) was administered via oral gavage to a final dose of 2.5 mg/10 g mouse for 5 consecutive days beginning at 5–8 weeks of age. Induced mice were group housed for at least 4 weeks prior to experiments.

Adult male and female mice (Cdh6$^{CreER}$+;Ai32 and Cdh6$^{CreER}$-;Ai32) aged 16–19 weeks were euthanized by isoflurane inhalation overdose in accordance with Flinders Animal Welfare Committee guidelines (ethics approval #4004). The protocol for animal euthanasia is approved by the National Health and Medical Research Council (NHMRC) Australian code for the care and use of animal for scientific

purposes (8th edition, 2013) and recommendations from the NHMRC Guidelines to promote the well-being of animals used for scientific purposes (2008).

## Dissection

Mice were culled by $CO_2$ and cervical dislocation. Small intestine and colon were removed and flushed with ice-cold PBS, then placed in a Sylgard-lined Petri dish with ice-cold PBS for further dissection.

### Wholemount preparations

Intestinal segments were prepared and fixed as in *Gomez Frittelli et al., 2023*. Briefly, each intestinal segment was opened along the mesentery border, pinned flat under light tension serosa-side up, and fixed in 4% PFA in PBS at 4°C with gentle rocking for 90 min. Segments were washed three times in PBS at 4°C for at least 10 min with gentle rocking. Muscularis was separated from the mucosa at one end of the segment with fine forceps for 2–3 mm, then pinned mucosa-side up in the dish. The mucosa was peeled away from the muscularis with fine forceps while the muscularis was gently held down in the dish with a cotton swab. For immunohistochemistry, segments were processed immediately or stored in PBS with 0.1% $NaN_3$ at 4°C until use. For RNAscope, muscularis segments were postfixed in 4% PFA in PBS at 4°C with gentle rocking overnight, then washed three times in PBS at 4°C with gentle rocking for at least 10 min each wash before use.

## Immunohistochemistry

Immunohistochemistry was performed as described previously (*Hamnett et al., 2022*). Briefly, muscularis wholemount tissue segments about 7 mm × 7 mm were incubated with primary antibodies in PBT (PBS with 1% BSA and 0.3% Triton X-100) at 4°C with gentle rocking overnight, then washed three times in PBT for at least 10 min each at room temperature with gentle shaking. Tissues were incubated in secondary antibodies in PBT for 2 hr with gentle shaking, washed twice in PBT, and twice in PBS, then mounted on Superfrost Plus slides with Fluoromount G medium (Southern Biotech). Primary antibodies: human anti-HuC/D (1:75k) (gift from Vanda Lennon); sheep anti-GFP (1:1k) (Biogenesis); rabbit anti-RFP (1:1k) (Rockland); rabbit anti-PGP9.5 (1:4k) (Abcam). Secondary antibodies: donkey anti-human Alexa Fluor (AF)-647 (1:500); donkey anti-sheep AF-488 (1:1k); donkey anti-rabbit AF-488 (1:1k); streptavidin AF-546 (1:500).

## RNAscope

In situ hybridization in combination with immunohistochemistry was performed on muscularis wholemount tissues using the RNAscope Multiplex Fluorescent V2 Assay kit with RNA-Protein Co-detection Ancillary Kit [ACD], according to the manufacturer's instructions with modifications as previously described (*Guyer et al., 2023*). Probes used were *Cdh6* (#519541), *Cdh8* (#485461), *Nmu* (#446831), *Calcb* (#425511), and *eGFP* (#400281).

## Confocal imaging

Images were acquired on a Leica SP8 confocal microscope using a ×20 (NA 0.75) oil objective at 1024 × 1024 pixel resolution. Tiled images (24–30 tiles) of z-stacks (2.5 µm between planes) were acquired and stitched together using the Navigator mode within LASX (Leica). Imaged regions were located away from the mesenteric border.

## Image analysis and quantification

Image analysis was performed using ImageJ/Fiji (NIH, Bethesda, MD, USA), as described previously (*Hamnett et al., 2022*). HuC/D images (z-stack individual planes) were blurred and thresholded, then maximally projected and total neurons counted using the Analyze Particles function. Cdh6*CreER*;Ai32 expression and RNAscope in situ hybridization and Cdh6*CreER*;Ai14 expression were counted manually. Cell tracing was performed in Imaris using Filament Tracer (Bitplane, Oxford Instruments).

## Electrophysiological recordings

Whole-cell patch-clamp electrophysiological recordings of Hb9:GFP+ neurons were performed according to *Osorio and Delmas, 2011*, with modifications for recording from the distal colon. The protocol is described in brief below.

## Tissue dissection and preparation

Mice aged 8–10 weeks were culled by $CO_2$ and cervical dislocation. The colon was removed and flushed with ice-cold oxygenated Krebs solution (118 mM NaCl, 4.8 mM KCl, 1 mM $NaH_2PO_4$, 25 mM $NaHCO_3$, 1.2 mM $MgCl_2$, 2.5 mM $CaCl_2$, and 11 mM glucose, supplemented with scopolamine [2 M] and nicardipine [6 μM]), then placed in a Sylgard-lined Petri dish with ice-cold oxygenated Krebs solution for further dissection. Krebs solution was changed out for fresh oxygenated solution every 5 min. Under a dissection microscope, the distal colon was pinned and the mucosa peeled away using fine forceps, leaving a few millimeters of mucosa along the edges of the tissue for pinning stability. The muscularis was then flipped over and re-pinned, serosa side up, and the longitudinal muscle carefully peeled away. The tissue was transferred to a custom 3D-printed recording chamber lined with a thin layer of clear Sylgard, and re-pinned under light tension, with the myenteric plexus facing up. The tissue was kept at 32°C and was continuously perfused with oxygenated Krebs solution. Hb9:GFP+ neurons were visually identified within a ganglion under epifluorescence illumination with a 455 nM LED (Thorlabs, M455L2) and a 470 (excitation)/525 (emission) nm wavelength filter set. A local perfusion of protease XIV (0.2% in Krebs) (Sigma, P5417) was applied on top of the targeted cell to digest any muscle fiber residue. A 1–2 MΩ pipet with a trimmed arm hair glued to the tip was used to brush and clean the surface of the ganglion. Further cleaning with 1 mg/mL collagenase (Worthington, CLS-4) 4 mg/mL dispase (Sigma, D4693) in Krebs solution was also performed to expose the GFP neuron for patching.

## Patching and recording

Patch pipettes (4–6 MΩ) pulled from borosilicate glass were filled with internal solution containing in mM: 144 K-gluconate, 3 $MgCl_2$, 0.5 EGTA, 10 HEPES, pH 7.2 (285/295 mOsm), and 2% biocytin (Millipore Sigma, B4261-100MG). Patch-clamp recordings were collected with a Multiclamp 700A (Molecular Devices) amplifier, a Digidata 1440 digitizer, and pClamp10.7 (Molecular Devices). Recordings were sampled and filtered at 10 kHz. Passive properties analysis was performed using pClamp10.7. Analysis of AP was performed using a custom MATLAB (MathWorks) software. All recordings were performed at 32°C. Membrane potentials were not corrected for liquid junction potential. Immediately after whole-cell configuration, the cell was maintained at –70 mV and a short voltage clamp membrane test protocol consisting of 20 times 600 ms, 10 mV depolarization steps was performed to assess cell health and recording conditions. Recordings were performed in Hb9:GFP+ colonic neurons with an access resistance less than 30 MΩ (16.69±2.63 MΩ). Next, the current clamp mode was used to measure RMP, input resistance (Rin), and APs stimulated. Membrane potential was not adjusted from resting potential, and cells were depolarized by 1 s current pulses in 10 pA increments until APs were triggered (rheobase). Finally, if the seal was still stable, a voltage clamp steady-state inactivation of T-current protocol was performed as previously described (*Huguenard and Prince, 1992*). In brief, a sequence of depolarization from –90 to –45 mV for 500 ms quickly followed by a depolarization to –40 mV for 200 ms. Tissues were then fixed and immunostained according to Wholemount preparations and Immunohistochemistry sections above.

## Mechanical recordings and optical stimulation

Optogenetic stimulation experiments were performed as previously described (*Hibberd et al., 2018a*). A 2.5 mm stainless-steel rod was inserted through the lumen of the colon and mounted in an organ bath (120*40*12 mm; L*W*H) located on a heated base. Krebs solution (35.5–36°C) superfused the bath (~5 mL/min). Smooth muscle force was recorded via four evenly spaced hooks in the colonic muscularis externa, each linked to an isometric force transducer (Grass FT03C) by suture thread. Initial base resting tension was set between 0.5 and 1.0 g. Preamplified signals (Biomedical Engineering, Flinders University) were digitized by a PowerLab 16/35 (ADInstruments, Bella Vista, NSW, Australia) and recorded using LabChart 7 software (ADInstruments) on iMac computer. Post hoc analysis of the mechanical recordings was done using LabChart 8 software on PC.

For optical stimulation during mechanical recordings in vitro, two LEDs (emitting 470 nm $\lambda$ photons; C470DA2432, Cree Inc, NC, USA) were used, driven by a variable power supply. The area of light emission from each LED was 240 μm × 320 μm (0.0768 mm²). To characterize LED function, light power density across a range of currents was measured 5 mm from the LED using a standard photodiode power sensor (S120C; Thorlabs, NJ, USA) and a power meter (Thorlabs, PM100USB). The

stimulator panel within LabChart software was used to set parameters and manually trigger LED pulse trains via the 10 V analogue output of the PowerLab and an ILD1 opto-isolator.

### Intraluminal pellet CMC recordings

To record proximal and distal colon CMCs separately (*Ishizawa, 1984*; *Hibberd et al., 2022*), full-length colon was bisected halfway between the caeco-colonic junction and terminal rectum, creating equal length proximal and distal colon preparations. Each preparation was suspended vertically on a stainless-steel holder inside a glass, water jacketed organ bath containing Krebs solution (*Figure 6J and K*). A 2.7 mm diameter synthetic pellet (polymethyl methacrylate, 'Perspex') was placed inside the gut lumen and linked by stainless-steel rod to a force transducer (MLT0420, ADInstruments), allowing measurement of both anterograde and retrograde propulsive forces on the pellet. Signals were amplified by bridge amplifier (FE224, ADInstruments), digitized at 1 kHz (PLCF1, ADInstruments) and recorded using LabChart 8 software.

ZD7288 (73777, Sigma-Aldrich) was dissolved in water as stock solution at 10 mM. Cesium chloride (C4036, Sigma-Aldrich) was dissolved in water as stock solution at 200 mM. Tetrodotoxin citrate (T-550, Alomone Labs) was dissolved in water as stock solution at 3 mM. Control, ZD7288, CsCl, and washout periods were at least 30 min; TTX was applied for at least 10 min.

### Statistical analysis

Statistical tests and graphical representation of data were performed using Prism 9 software (GraphPad). Statistical comparisons were performed using paired t tests (one-tailed, CMC intervals; two-tailed, peak amplitude, AUC, duration) and Welch's t test (marker colocalizations). Asterisks indicate significant differences.

### Study design

#### Sample size determination

Power analyses for previous similar experiments of neuronal marker cell counting determined that a sample size of n=5 was sufficient. For colonic motility experiments, sample sizes were based on 25% of measurement variance explained by the treatment effect (the minimum effect of interest) and a within group variance of 20% (effect size f=1.12). Repeated measures ANOVA with two groups with an alpha error probability of 5% and power of 95% gives a minimum sample size of five replicates in each group.

#### Randomization

Randomization was not relevant to our study, as all mice were allocated into experimental groups based on genotype.

#### Blinding

For optogenetic studies, experimenters were initially blinded to group allocation; however, it should be noted that overt functional responses to optogenetic stimulation unavoidably reveal group allocation once experiments are underway. For pharmacological experiments in control mice, it was not possible to blind the subject performing analysis of the mechanical force recordings.

#### Inclusion/exclusion criteria

All experiments were completed with replicates, and all replicates included in the data, with sample sizes detailed in the text and figure legends. Data was not excluded in this study.

## Acknowledgements

We thank the members of the Kaltschmidt laboratory for experimental advice and discussions, Vanda Lennon (Mayo Clinic) for the HuC/D primary antibody and Beatriz G Robinson for 3D printing of organ chambers for electrophysiology. National Institute of General Medical Sciences of the National Institutes of Health Award T32GM120007 (JGF), National Institutes of Health Grant R01 EY030138 (XD), National Health and Medical Research Council (NHMRC) project grant 1156416 (NJS), Australian Research Council (ARC) Discovery Project grant DP220100070 (NJS), NINDS 5R01NS34774 (JRH), Wu

Tsai Neurosciences Institute, Stanford University (JAK), Department of Neurosurgery, Stanford University (JAK), National Institutes of Health Grant R21 HD110950 (JAK), The Firmenich Foundation (JAK), The Carol and Eugene Ludwig Family Foundation (JAK), Stanford ADRC Developmental Project Grant (National Institutes of Health Grant P30AG066515) (JAK).

## Additional information

### Competing interests

Xin Duan, John R Huguenard: Reviewing editor, *eLife*. The other authors declare that no competing interests exist.

### Funding

| Funder | Grant reference number | Author |
| --- | --- | --- |
| National Institute of General Medical Sciences | T32GM120007 | Julieta Gomez-Frittelli |
| National Institutes of Health | R01 EY030138 | Xin Duan |
| National Health and Medical Research Council | 1156416 | Nick J Spencer |
| Australian Research Council | DP220100070 | Nick J Spencer |
| National Institute of Neurological Disorders and Stroke | 5R01NS34774 | John R Huguenard |
| Wu Tsai Neurosciences Institute, Stanford University | | Julia A Kaltschmidt |
| Department of Neurosurgery, Stanford University School of Medicine | | Julia A Kaltschmidt |
| National Institutes of Health | R21 HD110950 | Julia A Kaltschmidt |
| The Firmenich Foundation | | Julia A Kaltschmidt |
| The Carol and Eugene Ludwig Family Foundation | | Julia A Kaltschmidt |
| National Institutes of Health | P30AG066515 | Julia A Kaltschmidt |

The funders had no role in study design, data collection and interpretation, or the decision to submit the work for publication.

### Author contributions

Julieta Gomez-Frittelli, Conceptualization, Formal analysis, Investigation, Visualization, Methodology, Writing – original draft, Writing – review and editing; Gabrielle Frederique Devienne, Formal analysis, Investigation, Visualization, Methodology, Writing – original draft; Lee Travis, Melinda A Kyloh, Formal analysis, Investigation; Xin Duan, Methodology; Tim J Hibberd, Formal analysis, Investigation, Visualization, Methodology; Nick J Spencer, John R Huguenard, Resources, Supervision, Funding acquisition, Writing – review and editing; Julia A Kaltschmidt, Conceptualization, Resources, Supervision, Funding acquisition, Writing – review and editing

### Author ORCIDs

Julieta Gomez-Frittelli (ID) https://orcid.org/0000-0001-8859-9270
Gabrielle Frederique Devienne (ID) https://orcid.org/0000-0002-2204-8043
Xin Duan (ID) https://orcid.org/0000-0001-5260-8972

Nick J Spencer https://orcid.org/0000-0003-2190-5303
John R Huguenard https://orcid.org/0000-0002-6950-1191
Julia A Kaltschmidt https://orcid.org/0000-0002-2893-1793

### Ethics

All procedures conformed to the National Institutes of Health Guidelines for the Care and Use of Laboratory Animals and were approved by the Stanford University Administrative Panel on Laboratory Animal Care. Mice were group housed up to a maximum of five adults per cage. Food and water were provided ad libitum and mice were maintained on a 12:12 LD cycle. Male and female mice were used in all experiments. Adult male and female mice (Cdh6CreER+;Ai32 and Cdh6CreER-;Ai32) aged 16 to 19 weeks were euthanised by isoflurane inhalation overdose in accordance with Flinders Animal Welfare Committee guidelines (ethics approval #4004). The protocol for animal euthanasia is approved by the National Health and Medical Research Council (NHMRC) Australian code for the care and use of animal for scientific purposes (8th edition, 2013) and recommendations from the NHMRC Guidelines to promote the wellbeing of animals used for scientific purposes (2008).

Reviewer #1 (Public review): https://doi.org/10.7554/eLife.101043.3.sa1
Reviewer #2 (Public review): https://doi.org/10.7554/eLife.101043.3.sa2
Author response https://doi.org/10.7554/eLife.101043.3.sa3

---

## Additional files

### Supplementary files

MDAR checklist

### Data availability

Data have been uploaded and deposited in Zenodo (https://doi.org/10.5281/zenodo.14984617) and Dryad (https://doi.org/10.5061/dryad.66t1g1kbt).

The following datasets were generated:

| Author(s) | Year | Dataset title | Dataset URL | Database and Identifier |
|---|---|---|---|---|
| Gomez-Frittelli J, Devienne G, Travis L, Kyloh MA, Duan X, Hibberd TJ, Spencer NJ, Huguenard JR, Kaltschmidt JA | 2025 | Synaptic cell adhesion molecule Cdh6 identifies a class of sensory neurons with novel functions in colonic motility | https://zenodo.org/records/14984617 | Zenodo, 10.5281/zenodo.14984617 |
| Gomez-Frittelli J, Devienne G, Travis L, Duan X, Duan X, Kyloh M, Hibberd T, Spencer N, Huguenard J, Kaltschmidt J | 2026 | Data from: Synaptic cell adhesion molecule Cdh6 identifies a class of sensory neurons with novel functions in colonic motility | https://doi.org/10.5061/dryad.66t1g1kbt | Dryad Digital Repository, 10.5061/dryad.66t1g1kbt |

The following previously published dataset was used:

| Author(s) | Year | Dataset title | Dataset URL | Database and Identifier |
|---|---|---|---|---|
| Drokhlyansky E, Smillie CS, Wittenberghe NV, Ericsson M, Griffin GK, Eraslan G, Dionne D, Cuoco MS, Goder-Reiser MN, Sharova T, Kuksenko O, Aguirre AJ, Boland GM, Graham D, Rozenblatt-Rosen O, Xavier RJ, Regev A | 2020 | The human and mouse enteric nervous system at single cell resolution | https://singlecell.broadinstitute.org/single_cell/study/SCP1038/the-human-and-mouse-enteric-nervous-system-at-single-cell-resolution | Single Cell Portal, SCP1038 |

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
