## [Editor Report · eLife Assessment]

This **important** study characterizes the molecular signatures and function of a type of enteric neuron (IPAN) in the mouse colon, identifying molecular markers (Cdh6 and Cdh8) for these cells. A battery of **compelling** and comprehensive experimental findings suggests data from other species are likely translatable to mice, bridging the abundant literature from humans and other mammals into this experimentally tractable animal model. This work will be of interest to scientists studying the motor control of the colon and more generally the enteric neuromuscular system.

---

## [Referee Report · Reviewer #1 (Public review)]

Summary:

In their manuscript, Gomez-Frittelli and colleagues characterize the expression of cadherin6 (and -8) in colonic IPANs of mice. Moreover, they found that these cdh6-expressing IPANs are capable of initiating colonic motor complexes in the distal colon, but not proximal and midcolon. They support their claim by morphological, electrophysiological and optogenetic, and pharmacological experiments.

Strengths:

The work is very impressive and involves several genetic models and state-of-the-art physiological setups including respective controls. It is a very well-written manuscript that truly contributes to our understanding of GI-motility and its anatomical and physiological basis. The authors were able to convincingly answer their research questions with a wide range of methods without overselling their results.

Weaknesses:

The authors put quite some emphasis on stating that cdh6 is a synaptic protein (in the title and throughout the text), which interacts in a homophilic fashion. They deduct that cdh6 might be involved in IPAN-IPAN synapses (line 247ff.). However, Cdh6 does not only interact in synapses and is expressed by non-neuronal cells as well (see e.g., expression in the proximal tubuli of the kidney). Moreover, cdh6 does not only build homodimers, but also heterodimers with Chd9 as well as Cdh7, -10, and -14 (see e.g., Shimoyama et al. 2000, DOI: 10.1042/0264-6021:3490159). It would therefore be interesting to assess the expression pattern of cdh6-proteins using immunostainings in combination with synaptic markers to substantiate the authors' claim or at least add the possibility of cell-cell-interactions other than synapses to the discussion. Additionally, an immunostaining of cdh6 would confirm if the expression of tdTomato in smooth muscle cells of the cdh6-creERT model is valid or a leaky expression (false positive).

Comments on revisions:

The authors have updated their manuscript and have provided insights and discussions to my remarks.

---

## [Referee Report · Reviewer #2 (Public review)]

Summary:

Intrinsic primary afferent neurons are an interesting population of enteric neurons that transduce stimuli from the mucosa, initiate reflexive neurocircuitry involved in motor and secretory functions, and modulate gut immune responses. The morphology, neurochemical coding, and electrophysiological properties of these cells have been relatively well described in a long literature dating back to the late 1800's but questions remain regarding their roles in enteric neurocircuitry, potential subsets with unique functions, and contributions to disease. Here, the authors provide RNAscope, immunolabeling, electrophysiological, and organ function data characterizing IPANs in mice and suggest that Cdh6 is an additional marker of these cells.

Strengths:

This paper would likely be of interest to the enteric neuroscience community and increases information regarding the properties of IPANs in mice. These data are useful and suggest that prior data from studies of IPANs in other species are likely translatable to mice.

Weaknesses:

Major weaknesses:

(1) The novelty of this study is relatively limited. The main point of novelty suggests an additional marker of IPANs (Cdh6) that would add to the known list of markers for these cells. How useful this would be is unclear. Other main findings basically confirm that IPANs in mice display the same classical characteristics that have been known for many years from studies in guinea pigs, rats, mice and humans.

(2) Critical controls are needed to support the optogenetic experiments. Control experiments are needed to show that ChR2 expression (1) does not change the baseline properties of the neurons, (2) that stimulation with the chosen intensity of light elicits physiologically relevant responses in those neurons, and (3) that stimulation via ChR2 elicits comparable responses in IPANs in the different gut regions focused on here. These essential controls remain absent in the study and limit confidence in the data derived from this model.

(3) The motor effects observed in optogenetic experiments are difficult to understand in the absence of good controls for optogenetic control of the proposed neuron population (discussed above). It remains unclear how stimulating IPANs in the distal colon would generate retrograde CMCs while stimulating IPANs in the proximal colon did nothing. Key controls confirming that the optogentic stimulus was adequate, specific, and relevant are needed. In addition, better characterization of the Cdh6+ population of cells in both regions would be needed to understand the mechanisms underlying these effects.

(4) From the data shown, it is clear that expression driven by the Cdh6CreERT2 driver is not confined to IPANs. There is obviously expression of GFP and ChR2 in smooth muscle cells. This is a major limitation for the physiological experiments that attempt to use this model to specifically stimulate IPANs and assess changes in gut motor function. Better characterization of this model is needed and control experiments are necessary to assess whether functional ChR2 is expressed in cells beyond the proposed subtype of enteric IPANs.

(5) Some of the main conclusions of this study are overstated and claims of priority are made that are not true. For example, the authors state on lines 27-28 of the abstract that their findings provide the "first demonstration of selective activation of a single neurochemical and functional class of enteric neurons". This is certainly not true since Gould et al (AJP-GIL 2019) expressed ChR2 in nitrergic enteric neurons and showed that activating those cells disrupted CMC activity. In fact, prior work by the authors themselves (Hibberd et al Gastro 2018) showed that activating calretinin neurons with ChR2 evoked motor responses. Work by other groups has used chemogenetics and optogenetics to show effects of activating multiple other classes of neurons in the gut.

(6) The electrophysiological characterization of mouse IPANs is useful but is limited to a small subset of Cdh6+ neurons in the distal colon myenteric plexus. Therefore, it remains unclear how well the properties reported here might reflect those of other Cdh6+ IPANs in the same or different regions. Similarly, blocking IH with ZD7288 affects all IPANs and does not add specific information regarding the role of the proposed Cdh6+ subtype.

(7) The submucosal plexus (SMP) also contains enteric IPANs and these were not included in the analysis of Cdh6 expression. Whether or not the proposed IPAN marker Cdh6 would be useful for identifying or targeting those cells remains unclear.

[Editor's note: The Reviewing Editor considers that further controls requested from the reviewers have largely been provided already in prior publications by other groups, as they concern specifically tools published years ago but in a different tissue context. Hence the methodology used to deliver the results reported here fall within the standard practices in the field. The comprehensive, multi-technique approach to the results is compelling in and of itself, and ought to suffice, rendering this work reproducible and therefore a basis for further research.]

---

## [Author Response]

The following is the authors’ response to the original reviews.

**Reviewer #1 (Public review):**
Summary:In their manuscript, Gomez-Frittelli and colleagues characterize the expression of cadherin6 (and -8) in colonic IPANs of mice. Moreover, they found that these cdh6-expressing IPANs are capable of initiating colonic motor complexes in the distal colon, but not proximal and midcolon. They support their claim by morphological, electrophysiological, optogenetic, and pharmacological experiments.Strengths:The work is very impressive and involves several genetic models and state-of-the-art physiological setups including respective controls. It is a very well-written manuscript that truly contributes to our understanding of GI-motility and its anatomical and physiological basis. The authors were able to convincingly answer their research questions with a wide range of methods without overselling their results.

We greatly appreciate the reviewer’s time, careful reading and support of our study.

Weaknesses:The authors put quite some emphasis on stating that cdh6 is a synaptic protein (in the title and throughout the text), which interacts in a homophilic fashion. They deduct that cdh6 might be involved in IPAN-IPAN synapses (line 247ff.). However, Cdh6 does not only interact in synapses and is expressed by non-neuronal cells as well (see e.g., expression in the proximal tubuli of the kidney). Moreover, cdh6 does not only build homodimers, but also heterodimers with Chd9 as well as Cdh7, -10, and -14 (see e.g., Shimoyama et al. 2000, DOI: 10.1042/02646021:3490159). It would therefore be interesting to assess the expression pattern of cdh6proteins using immunostainings in combination with synaptic markers to substantiate the authors' claim or at least add the possibility of cell-cell-interactions other than synapses to the discussion. Additionally, an immunostaining of cdh6 would confirm if the expression of tdTomato in smooth muscle cells of the cdh6-creERT model is valid or a leaky expression (false positive).

We agree with the reviewer that Cdh6 could be mediating some other cell-cell interaction besides synapses between IPANs, and we noted it in the discussion. Cdh6 primarily forms homodimers but, as the reviewer points out, has been known to also form heterodimers with some other cadherins. We performed RNAscope in the colonic myenteric plexus with Cdh7 and found no expression (data not shown). Cdh10 is suggested to have very low expression (Drokhlyansky et al., 2020), possibly in putative secretomotor vasodilator neurons, and Cdh14 has not been assayed in any RNAseq screens. We attempted to visualize Cdh6 protein via antibody staining (Duan et al., 2018) but our efforts did not result in sufficient signal or resolution to identify synapses in the ENS, which remain broadly challenging to assay. Similarly, immunostaining with Cdh6 antibody was unable to confirm Cdh6 protein in tdT-expressing muscle cells, or by RNAscope. We have addressed these caveats in the discussion section.

(1) E. Drokhlyansky, C. S. Smillie, N. V. Wittenberghe, M. Ericsson, G. K. Griffin, G. Eraslan, D. Dionne, M. S. Cuoco, M. N. Goder-Reiser, T. Sharova, O. Kuksenko, A. J. Aguirre, G. M. Boland, D. Graham, O. Rozenblatt-Rosen, R. J. Xavier, A. Regev, The Human and Mouse Enteric Nervous System at Single-Cell Resolution. Cell 182, 1606-1622.e23 (2020).

(2) X. Duan, A. Krishnaswamy, M. A. Laboulaye, J. Liu, Y.-R. Peng, M. Yamagata, K. Toma, J. R. Sanes, Cadherin Combinations Recruit Dendrites of Distinct Retinal Neurons to a Shared Interneuronal Scaffold. Neuron 99, 1145-1154.e6 (2018).

**Reviewer #2 (Public review):**
Summary:Intrinsic primary afferent neurons are an interesting population of enteric neurons that transduce stimuli from the mucosa, initiate reflexive neurocircuitry involved in motor and secretory functions, and modulate gut immune responses. The morphology, neurochemical coding, and electrophysiological properties of these cells have been relatively well described in a long literature dating back to the late 1800's but questions remain regarding their roles in enteric neurocircuitry, potential subsets with unique functions, and contributions to disease. Here, the authors provide RNAscope, immunolabeling, electrophysiological, and organ function data characterizing IPANs in mice and suggest that Cdh6 is an additional marker of these cells.Strengths:This paper would likely be of interest to a focused enteric neuroscience audience and increase information regarding the properties of IPANs in mice. These data are useful and suggest that prior data from studies of IPANs in other species are likely translatable to mice.

We appreciate the reviewer’s support of our study and insightful critiques for its improvement.

Weaknesses:The advance presented here beyond what is already known is minimal. Some of the core conclusions are overstated and there are multiple other major issues that limit enthusiasm. Key control experiments are lacking and data do not specifically address the properties of the proposed Cdh6+ population.Major weaknesses:(1) The novelty of this study is relatively low. The main point of novelty suggests an additional marker of IPANs (Cdh6) that would add to the known list of markers for these cells. How useful this would be is unclear. Other main findings basically confirm that IPANs in mice display the same classical characteristics that have been known for many years from studies in guinea pigs, rats, mice and humans.

We appreciate the already existing markers for IPANs in the ENS and the existing literature characterizing these neurons. The primary intent of this study was to use these well-established characteristics of IPANs in both mice and other species to characterize Cdh6-expressing neurons in the mouse myenteric plexus and confirm their classification as IPANs.

(2) Some of the main conclusions of this study are overstated and claims of priority are made that are not true. For example, the authors state in lines 27-28 of the abstract that their findings provide the "first demonstration of selective activation of a single neurochemical and functional class of enteric neurons". This is certainly not true since Gould et al (AJP-GIL 2019) expressed ChR2 in nitrergic enteric neurons and showed that activating those cells disrupted CMC activity. In fact, prior work by the authors themselves (Hibberd et al., Gastro 2018) showed that activating calretinin neurons with ChR2 evoked motor responses. Work by other groups has used chemogenetics and optogenetics to show the effects of activating multiple other classes of neurons in the gut.

We thank the reviewer for bringing up this important point and apologize if our wording was not clear. Whilst single neurochemical classes of enteric neurons have been manipulated to alter gut functions, all such instances to date do not represent manipulation of a single functional class of enteric neurons. In the given examples, multiple functional classes are activated utilizing the same neurotransmitter, as NOS and calretinin are each expressed to varying degrees across putative motor neurons, interneurons and IPANs. In contrast, Chd6 is restricted to IPANs and therefore this study is the first optogenetic investigation of enteric neurons from a single putative functional class. Our abstract and discussion emphasizes this point and differentiates this study from those previous.

(3) Critical controls are needed to support the optogenetic experiments. Control experiments are needed to show that ChR2 expression (a) does not change the baseline properties of the neurons, (b) that stimulation with the chosen intensity of light elicits physiologically relevant responses in those neurons, and (c) that stimulation via ChR2 elicits comparable responses in IPANs in the different gut regions focused on here.

We completely agree controls are essential. However, our paper is not the first to express ChR2 in enteric neurons. Authors of our paper have shown in Hibberd *et al.* 2018 that expression of ChR2 in a heterogeneous population of myenteric neurons did not change network properties of the myenteric plexus. This was demonstrated in the lack of change in control CMC characteristics in mice expressing ChR2 under basal conditions (without blue light exposure). Regarding question (b), that it should be shown that stimulation with the chosen intensity of light elicits physiologically relevant responses in those neurons. We show the restricted expression of ChR2 in IPANs and that motor responses (to blue light) are blocked by selective nerve conduction blockade.

Regarding question (c), that our study should demonstrate that stimulation via ChR2 elicits comparable responses in IPANs in the different gut regions. We would not expect each region of the gut to behave comparably. This is because the different gut regions (i.e. proximal, mid, distal) are very different anatomically, as is anatomy of the myenteric plexus and myenteric ganglia between each region, including the density of IPANs within each ganglia, in addition to the presence of different patterns of electrical and mechanical activity [Spencer *et al.,* 2020]. Hence, it is difficult to expect that between regions stimulation of ChR2 should induce similar physiological responses. The motor output we record in our study (CMCs) is a unified motor program that involves the temporal coordination of hundreds of thousands of enteric neurons and a complex neural circuit that we have previously characterized [Spencer *et al.*, 2018]. But, never has any study until now been able to selectively stimulate a single functional class of enteric neurons (with light) to avoid indiscriminate activation of other classes of neurons.

(1) T. J. Hibberd, J. Feng, J. Luo, P. Yang, V. K. Samineni, R. W. Gereau, N. Kelley, H. Hu, N. J. Spencer, Optogenetic Induction of Colonic Motility in Mice. Gastroenterology 155, 514-528.e6 (2018).

(2) N. J. Spencer, L. Travis, L. Wiklendt, T. J. Hibberd, M. Costa, P. Dinning, H. Hu, Diversity of neurogenic smooth muscle electrical rhythmicity in mouse proximal colon. American Journal of Physiology-Gastrointestinal and Liver Physiology 318, G244–G253 (2020).

(3) N. J. Spencer, T. J. Hibberd, L. Travis, L. Wiklendt, M. Costa, H. Hu, S. J. Brookes, D. A. Wattchow, P. G. Dinning, D. J. Keating, J. Sorensen, Identification of a Rhythmic Firing Pattern in the Enteric Nervous System That Generates Rhythmic Electrical Activity in Smooth Muscle. The Journal of Neuroscience 38, 5507–5522 (2018).

(4) The electrophysiological characterization of mouse IPANs is useful but this is a basic characterization of any IPAN and really says nothing specifically about Cdh6+ neurons. The electrophysiological characterization was also only done in a small fraction of colonic IPANs, and it is not clear if these represent cell properties in the distal colon or proximal colon, and whether these properties might be extrapolated to IPANs in the different regions. Similarly, blocking IH with ZD7288 affects all IPANs and does not add specific information regarding the role of the proposed Cdh6+ subtype.

Our electrophysiological characterization was guided to be within a subset of Cdh6+ neurons by Hb9:GFP expression. As in the prior comment (1) above, we used these experiments to confirm classification of Cdh6+ (Hb9:GFP+) neurons in the distal colon as IPANs. We have clarified in the results and methods that these experiments were performed in the distal colon and agree that we cannot extrapolate that these properties are also representative of IPANs in the proximal colon. We apologize that this was confusing. Finally, we agree with the reviewer that ZD7288 affects all IPANs in the ENS and have clarified this in the text.

(5) Why SMP IPANs were not included in the analysis of Cdh6 expression is a little puzzling. IPANs are present in the SMP of the small intestine and colon, and it would be useful to know if this proposed marker is also present in these cells.

We agree with the reviewer. In addition to characterizing Cdh6 in the myenteric plexus, it would be interesting to query if sensory neurons located within the SMP also express Cdh6. Our preliminary data (n=2) show ~6-12% tdT/Hu neurons in Cdh6-tdT ileum and colon (data not shown). We have added a sentence to the discussion.

(6) The emphasis on IH being a rhythmicity indicator seems a bit premature. There is no evidence to suggest that IH and IT are rhythm-generating currents in the ENS.

Regarding the statement there is no evidence to suggest that IH and IT are rhythm-generating currents in the ENS. We agree with the reviewer that evidence of rhythm generation by IH and IT in the ENS has not been explicitly confirmed. We are confident the reviewer agrees that an absence of evidence is not evidence of absence, although the presence of IH has been well described in enteric neurons. We have modified the text in the results to indicate more clearly that IH and IT are known to participate in rhythm generation in thalamocortical circuits, though their roles in the ENS remain unknown. Our discussion of the potential role of IH or IT in rhythm generation or oscillatory firing of the ENS is constrained to speculation in the discussion section of the text.

(7) As the authors point out in the introduction and discuss later on, Type II Cadherins such as Cdh6 bind homophillically to the same cadherin at both pre- and post-synapse. The apparent enrichment of Cdh6 in IPANs would suggest extensive expression in synaptic terminals that would also suggest extensive IPAN-IPAN connections unless other subtypes of neurons express this protein. Such synaptic connections are not typical of IPANs and raise the question of whether or not IPANs actually express the functional protein and if so, what might be its role. Not having this information limits the usefulness of this as a proposed marker.

We agree with the reviewer that the proposed IPAN-IPAN connection is novel although it has been proposed before (Kunze et al., 1993). As detailed in our response to Reviewer #1, we attempted to confirm Cdh6 protein expression, but were unsuccessful, due to insufficient signal and resolution. We therefore discuss potential IPAN interconnectivity in the discussion, in the context of contrasting literature.

(1) W. A. A. Kunze, J. B. Furness, J. C. Bornstein, Simultaneous intracellular recordings from enteric neurons reveal that myenteric ah neurons transmit via slow excitatory postsynaptic potentials. Neuroscience 55, 685–694 (1993).

(8) Experiments shown in Figures 6J and K use a tethered pellet to drive motor responses. By definition, these are not CMCs as stated by the authors.

The reviewer makes a valid criticism as to the terminology, since tethered pellet experiments do not record propagation. We believe the periodic bouts of propulsive force on the pellet is triggered by the same activity underlying the CMC. In our experience, these activities have similar periodicity, force and identical pharmacological properties. Consistent with this, we also tested full colons (n = 2) set up for typical CMC recordings by multiple force transducers, finding that CMCs were abolished by ZD7288, similar to fixed pellet recordings (data not shown).

(9) The data from the optogenetic experiments are difficult to understand. How would stimulating IPANs in the distal colon generate retrograde CMCs and stimulating IPANs in the proximal colon do nothing? Additional characterization of the Cdh6+ population of cells is needed to understand the mechanisms underlying these effects.

We agree that the different optogenetic responses in the proximal and distal colon are challenging to interpret, but perhaps not surprising in the wider context. It is not only possible that the different optogenetic responses in this study reflect regional differences in the Chd6+ neuronal populations, but also differences in neural circuits within these gut regions. A study some time ago by the authors showed that electrical stimulation of the proximal mouse colon was unable to evoke a retrograde (aborally) propagating CMC (Spencer, Bywater, 2002), but stimulation of the distal colon was readily able to. We concluded that at the oral lesion site there is a preferential bias of descending inhibitory nerve projections, since the ascending excitatory pathways have been cut off. In contrast, stimulation of the distal colon was readily able to activate an ascending excitatory neural pathway, and hence induce the complex CMC circuits required to generate an orally propagating CMC. Indeed, other recent studies have added to a growing body of evidence for significant differences in the behaviors and neural circuits of the two regions (Li et al., 2019, Costa et al., 2021a, Costa et al., 2021b, Nestor-Kalinoski et al., 2022). We have expanded this discussion.

(1) N. J. Spencer, R. A. Bywater, Enteric nerve stimulation evokes a premature colonic migrating motor complex in mouse. Neurogastroenterology & Motility 14, 657–665 (2002).

(2) Li Z, Hao MM, Van den Haute C, Baekelandt V, Boesmans W, Vanden Berghe P, Regional complexity in enteric neuron wiring reflects diversity of motility patterns in the mouse large intestine. Elife 8:e42914 (2019).

(3) Costa M, Keightley LJ, Hibberd TJ, Wiklendt L, Dinning PG, Brookes SJ, Spencer NJ, Motor patterns in the proximal and distal mouse colon which underlie formation and propulsion of feces. Neurogastroenterology & Motility e14098 (2021a).

(4) Costa M, Keightley LJ, Hibberd TJ, Wiklendt L, Smolilo DJ, Dinning PG, Brookes SJ, Spencer NJ, Characterization of alternating neurogenic motor patterns in mouse colon. Neurogastroenterology & Motility 33:e14047 (2021b).

(5) Nestor-Kalinoski A, Smith-Edwards KM, Meerschaert K, Margiotta JF, Rajwa B, Davis BM, Howard MJ, Unique Neural Circuit Connectivity of Mouse Proximal, Middle, and Distal Colon Defines Regional Colonic Motor Patterns. Cellular and Molecular Gastroenterology and Hepatology 13:309-337.e303 (2022).

**Recommendations for the Authors:**

**Reviewer #1 (Recommendations for the authors):**
As mentioned above, immunolocalization of cdh6 would be helpful to substantiate the claims regarding IPAN-IPAN synapses.

As mentioned in our response to both reviewers’ public reviews, we attempted to visualize Cdh6 protein via antibody staining (Duan et al., 2018), but our efforts did not result in sufficient signal or resolution to identify Cdh6+ synapses.

(1) X. Duan, A. Krishnaswamy, M. A. Laboulaye, J. Liu, Y.-R. Peng, M. Yamagata, K. Toma, J. R. Sanes, Cadherin Combinations Recruit Dendrites of Distinct Retinal Neurons to a Shared Interneuronal Scaffold. Neuron 99, 1145-1154.e6 (2018).

**Reviewer #2 (Recommendations for the authors):**
(1) The authors repeatedly refer to IPANs as "sensory" neurons (e.g. in title, abstract, and introduction) but there is some debate regarding whether these cells are truly "sensory" because the information they convey never reaches sensory perception. This is why they have classically been referred to as intrinsic primary afferent (IPAN) neurons. It would be more appropriate to stick with this terminology unless the authors have compelling data showing that information detected by IPANs reaches the sensory cortex.

We thank the reviewer for their comment, but respectfully disagree. The term “sensory neuron” is well established in the ENS. The first definitive proof that “sensory neurons” exist in the ENS was published in Kunze et al., 1995. We note that this paper did not use the word “IPAN” but used the term “sensory neuron”. Furthermore, mechanosensory neurons were published in Spencer and Smith (2004).

Regarding the reviewer’s comment that the authors would need compelling data showing that information detected by IPANs reaches the sensory cortex before the term “sensory neuron” should be valid, it is important to note that many sensory neurons do not provide direct information to the cortex.

(1) W. A. A. Kunze, J. C. Bornstein, J. B. Furness, Identification of sensory nerve cells in a peripheral organ (the intestine) of a mammal. Neuroscience 66, 1–4 (1995).

(2) N. J. Spencer, T. K. Smith, Mechanosensory S-neurons rather than AH-neurons appear to generate a rhythmic motor pattern in guinea-pig distal colon. The Journal of Physiology 558, 577–596 (2004).

(2) Important information regarding the gut region shown and other details are absent from many figure legends.

We apologize for this omission. We have updated the figure legends to include information on gut regions.